# Dementia-Free Life Expectancy among People over 60 Years Old by Sex, Urban and Rural Areas in Jiangxi Province, China

**DOI:** 10.3390/ijerph17165665

**Published:** 2020-08-05

**Authors:** Yuhang Wu, Huilie Zheng, Zhitao Liu, Shengwei Wang, Yong Liu, Songbo Hu

**Affiliations:** Jiangxi Province Key Laboratory of Preventive Medicine, School of Public Health, Nanchang University, Nanchang 330006, China; 411437819016@email.ncu.edu.cn (Y.W.); zhenghuilie@ncu.edu.cn (H.Z.); 401440319007@email.ncu.edu.cn (Z.L.); 411437818006@email.ncu.edu.cn (S.W.); ly070310@ncu.edu.cn (Y.L.)

**Keywords:** dementia, dementia-free life expectancy, elderly Chinese, urban–rural areas

## Abstract

Objective: To estimate and compare the dementia-free life expectancy (DemFLE) and age trends of the population over 60 in 2018 in Jiangxi Province, China, by sex and urban–rural areas. Methods: Based on the Summary of Health Statistics of Jiangxi Province in 2018 and the Sixth National Health Service survey of Jiangxi Province, the model life table is used to estimate the age-specific mortality rate by sex and urban–rural areas. DemFLE and its ratio to life expectancy (LE) were calculated using the Sullivan method. Results: In 2018, the DemFLE at age 60 was 18.48 years for men and 21.31 years for women, accounting for 96.62% and 96.67% of their LE. LE and DemFLE were higher for those in urban areas than in rural areas, except for men aged 90 and above; higher in women than in men, except for people in rural areas aged 90 and above. In urban areas, DemFLE/LE was higher for women than for men; the opposite was observed in rural areas. Urban women had a higher DemFLE/LE than rural women did, urban men had a lower DemFLE/LE than rural men did. Conclusions: With increased LE, DemFLE also increases, but with older age and over time, DemFLE/LE gradually decreases. The effect of dementia on elderly adults becomes more serious. It is necessary for the government to implement a series of prevention strategies to improve the quality of life and health awareness of the elderly. Elderly urban men and elderly rural women need more attention and health care.

## 1. Introduction

As early as 1980, Kramer [1] predicted that future pandemics would be mental disorders and related chronic diseases and disabilities. The burden of death and disability caused by neurological diseases such as dementia is increasingly recognized in global public health [2]. According to a systematic analysis of the Global Burden of Disease study [3], between 1950 and 2017, life expectancy increased from 48.1 years to 70.5 years for men and from 52.9 years to 75.6 years for women. However, there is a cost to extending life expectancy, as it can lead to an increasing number of people suffering from dementia [4].

According to the World Alzheimer Report 2018 [5], 50 million people were suffering from dementia in the world in 2018; this figure is expected to increase to 82 million in 2030. Currently, there is an average of one new dementia patient every three seconds in the world. The cost of dementia is estimated at $1 trillion. Dementia is a major economic burden on society, similar to that of heart disease and cancer [6]. In China, the media and general population tend to pay more attention to diseases with high mortality, such as cancer and cardiovascular diseases [7]. Chinese society has not given enough attention and help to those who “live with diseases”.

Dementia is a clinical syndrome and a general term. It is used to describe the chronic and progressive decline of the brain’s cognitive and functional abilities, thus interfering with normal daily life and social activities [8,9]. In addition, many patients with dementia often have other comorbidities, including depression [10,11], anxiety [12], and stroke [13,14]. This not only seriously reduces the quality of life of dementia patients but also leads to the heavy burden of formal care [15] and informal care [16].

Formal care is mainly provided by hospital nurses, who need to have the ability to treat and care for dementia patients [17]. In China, the heavy workload, low salary, and poor working conditions of nursing staff, combined with the traditional belief in the superiority of medical care over nursing care in the Chinese population, have reduced the social status of nursing staff, resulting in an extreme shortage of nurses in China [18]. Informal care is unpaid care provided by family and friends in the form of activities of daily living (ADLs) [16]. Families and friends of dementia patients form a very important support network in Chinese society [19].

Before we started our research, some scholars conducted research on dementia-free life expectancy (DemFLE) for men and women in Australia [20,21], Belgium [22], Canada [23,24], France [25,26], Japan [27], the Netherlands [28], the United Kingdom [29,30,31], the United States [32,33,34,35], Brazil [36] and Hong Kong [37]. There are no studies on DemFLE in mainland China. In addition, previous studies have reported only sex-specific differences in dementia-free life expectancy. However, because of evidence [38,39] that both sex and urban–rural area differences are critical in chronic noncommunicable diseases (NCDs) prevalence, our analysis of urban–rural differences is sex-specific.

In this study, the DemFLE and its ratio to life expectancy (LE) of specific age groups were calculated in Jiangxi Province by sex and urban–rural areas to investigate changes in the number of healthy years and in the quality of life of people of different age groups, different sexes and different areas. We use the ratio of life expectancy without dementia to life expectancy (DemFLE/LE) as an indicator to evaluate the quality of life of the elderly in Jiangxi Province, China, and to a certain extent can reflect the impact of dementia on the quality of life of the elderly. Therefore, the government, the health sector and the public can respond in a timely manner.

This is the first study of life expectancy among individuals without dementia in mainland China. We hope this study will provide a reference point for similar studies elsewhere with the aim of establishing a basis for both domestic and international comparisons.

## 2. Materials and Methods

Our study mainly requires the following two kinds of raw data:

The Summary of Health Statistics of Jiangxi Province [40], compiled by the Health Commission of Jiangxi Province, records data on the development of health undertakings in the whole province. The Health Commission of Jiangxi Province performs relevant statistics and records them in this book. According to these data, the book of the corresponding year will be issued separately every year. We used information on the infant mortality rate (IMR) and the under-five mortality rate (U5MR) in 2018 by gender and urban–rural areas, and the data come from the maternal and child monitoring system. On the basis of these data, we used the China model life table method [41] to estimate age-specific mortality rates and life expectancy by sex and urban and rural areas in Jiangxi Province in 2018. This method was constructed using the Murray model life table method [42] based on the census and population sampling data of China.

The dementia prevalence data used in this study were from the 6th National Health Service Survey in Jiangxi Province, China, which was a national sample survey. The sample was selected by multistage stratified cluster random sampling. Donghu District, Zhanggong District, Yuanzhou District, Shanggao County, Gao’an County and Panyang County were selected as the sample counties (cities, districts) representing the overall situation of urban and rural areas in Jiangxi Province. The survey was based on family health inquiry, and the data were collected by door-to-door inquiry. All members of the survey households were interviewed according to questionnaires administered by trained and qualified investigators. The investigators comprised health personnel from county (city, district) health institutions and township health centres or community health service centres. They had specific professional knowledge. The survey data were entered using the data entry software provided by the Statistical Information Center of the Ministry of Health. To ensure the quality of data entry, the data from the family health inquiry survey were entered twice. We obtained a total of 2711 valid responses from 10,123 participants (Aged 60 and above: 2784) to answer the question of whether they had memory loss or dementia; these participants were included in the study. The question described the participants’ dementia status, if they answered “yes”, they were considered to have dementia, and if they answered “no”, they were not considered to have dementia. All patients were at least 60 years old. The response rate for the dementia survey among people in their 60s was about 97%. Similar studies in other countries had reported the response rate for the dementia survey, such as the Netherlands (73%) [28], Australia (83%) [21] and Japan (69%) [27]. The reason for the high response rate of the survey was that the survey was only based on the answers to the questionnaire without physical and mental examinations, so that few people were lost to follow-up or rejected. Sullivan’s method [43] is the most common method for calculating healthy life expectancy. The life table is used to estimate the life expectancy of different age groups, and life expectancy is divided into healthy life expectancy and unhealthy life expectancy. To compare the difference between males and females and between individuals from urban and rural areas, we also calculated DemFLE/LE, which is the ratio of life expectancy without dementia to life expectancy. The standard error of the dementia rate was used to calculate the standard error of the DemFLE to calculate the corresponding 95% confidence intervals (CI) [44].

## 3. Results

Table 1 shows the results for those over 60 years old in great detail. Figure 1 and Figure 2 also present the trends of LE, DemLE, and DemFLE and the percentage of total life expectancy that one can expect to live free of dementia (DemFLE/LE).

The LE for those aged 60 in Jiangxi Province, China, is 19.13 years for men and 22.03 years for women. Men aged 60 years can expect to live 18.48 years without dementia, which accounts for 96.62% of LE. For women, these figures were 21.31 years and 96.76%, respectively, both of which were higher than those for men. At the age of 90, men have an LE of 3.67 years and a DemFLE of 2.61 (2.32, 2.90) years. The LE for women is 4.05 years, and they can expect to live 2.63 (2.24, 3.01) years without dementia. There is no statistical significance in DemFLE between men and women aged 90 and above. However, DemFLE/LE decreases to 71.08% for men and 64.87% for women at the age of 90.

Table 2 divides the total sample into urban and rural residents and describes their results. At the age of 60 years, LE for urban men is 20.07 years, and they can expect to live 19.31 years without dementia. For rural men, these figures are 18.96 years and 18.35 years, respectively. LE and DemFLE are 23.10 years and 22.38 years for urban women and 21.84 years and 21.02 years for rural women, respectively. Regardless of sex, LE and DemFLE for residents of urban areas are higher than for residents of rural areas, except for men aged 90 and above. Regardless of residential area, LE and DemFLE are higher for women than for men, except for people in rural areas aged 90 and above.

Figure 3 depicts the trends of dementia-free life expectancy as a proportion of life expectancy for men and women in urban and rural areas. At 60, life expectancy without dementia represented 96.22% of the life expectancy for urban men, 96.90% of that for urban women, 96.80% for rural men, and 96.28% for rural women. At the age of 90 and above, the proportion decreases to 65.94% for urban men, 67.60% for urban women, 78.89% for rural men, and 58.26% for rural women. Dementia-free life expectancy as a proportion of life expectancy for urban women aged 60 is always slightly higher than that for urban men of the same age; that for rural men is always higher than that for urban men; that for urban women is higher than that for rural women and that for rural men is higher than that for rural women. However, at the age of 80, life expectancy without dementia for rural women accounts for 91.76% of life expectancy, which is higher than that for urban women and rural men in the same age group.

Figure 4 presents the absolute changes (urban–rural) in life expectancy and dementia-free life expectancy between residents of urban and rural areas. Comparing urban and rural residents, life expectancy at age 60 increased by 1.11 years for men and 1.27 years for women. At a much older age (90 years old), the increase in life expectancy is only 0.29 years for men and 0.40 years for women, which is lower than that at younger ages. The absolute changes in dementia-free life expectancy for those aged 60 is 0.96 years for men and 1.36 years for women. The increase in DemFLE is smaller than the LE gain for men aged 60; the opposite is true for women aged 60, suggesting an absolute expansion of the number of years lived with dementia by urban men and rural women. Remarkably, at the age of 90 and above, these changes decreased by 0.28 years for men and increased by 0.64 years for women.

## 4. Discussion

As in previous studies, [20,21,22,23,24,25,26,27,28,29,30,31,32,33,34,35,36,37] we first analysed the DemFLE and its proportion among people over 60 years of age in Jiangxi Province by sex differences. Compared with previous studies, our study showed similarities and differences in LE, DemFLE and DemFLE/LE [20,21,22,23,24,25,26,27,28,29,30,31,32,33,34,35,36,37]. The difference between China and other countries in the values of LE, DemLE and DemFLE may be caused by the influences of sampling error, different sampling methods, different diagnostic criteria and differences between Eastern and Western countries. In addition, the length of time examined in other studies was very different from that in this study, so there were major differences in economic, cultural and other factors. With older age and over time, LE and DemFLE both increased, DemLE slightly increased, while DemFLE/LE gradually decreased. In addition, at every age, women had a longer LE and DemFLE than men did, and women experienced dementia for a longer period of time. It is worth mentioning that in this study, it is not yet believed that there is a significant difference in life expectancy without dementia between men and women aged 90 and above. This may be due to the insufficient sample size (men: 32, women: 29) of the surveyed aged 90 and above. The reason for this gender difference was the higher incidence of dementia in older women and the lower mortality rate for women than for men [45,46]. However, our study also found that before age 85, women had a slightly higher proportion of life expectancy without dementia than men did; in other words, men and women suffered roughly the same amount of morbidity. After age 90, the proportion of women (64.87%) was lower than that of men (71.08%); this finding can be attributed to the combination of morbidity and mortality [27]. Therefore, if the length of life were the sole focus, women had a clear advantage. However, with respect to the definition of health, the progress women have made in pursuing a high quality of life is not as extensive as we think. In fact, women exhibit a combination of good and poor health [21].

We conducted a discussion based on urban and rural areas to rationally guide the allocation of limited health resources. DemFLE showed a decreasing trend with increasing age, which was parallel to the decreasing trend of LE [25]. This was similar to the above trend. Before the age of 75, the difference in DemFLE/LE between urban and rural areas was less than 1%. By the age of 80, the proportion of rural women was 91.76%, which was higher than that of other groups (urban men: 89.10%, urban women: 89.63%, rural men: 90.78%). We use linear regression analysis for each sex to test the age trend of the ratio of DemFLE/LE. We did the same work for urban–rural areas, respectively. The results show that all of the *p* values were less than 0.05, which indicates that with the increase of age, DemFLE/LE shows a significant downward trend. With increasing age, urban men and rural women had a poorer quality of life because of dementia. The difference between men and women in rural areas was similar to that in the whole population, but the difference was more obvious. What merits our attention is that urban women not only have a longer life expectancy and a longer dementia-free life expectancy than urban men have, but are also less affected by dementia. We suspected that differences in lifestyle between men and women may lead to hypertension, hyperlipidaemia, stroke and other diseases in individuals in urban areas [47].

With increasing age, the absolute change in LE for men in urban and rural areas maintained positive growth, but the difference in LE between individuals in urban and rural areas gradually decreased. Although the same was true for absolute changes in DemFLE (except for 90 years), the increase in DemFLE was smaller than the LE gain. Urban men had more time with dementia than did rural men. Figure 4 shows “negative growth” by age 90, and the absolute values of DemFLE of rural men were higher than those of urban men. In China, rural men have more opportunities to engage in physical labour, and regular exercise can reduce the risk of dementia [48]. In addition, we also assumed that if an elderly man in a rural area had dementia, he was likely to die before age 90. In contrast, urban men spent a considerable proportion of their remaining life with dementia, even if they lived past age 90. On the other hand, because we calculated the prevalence of dementia in rural men aged 90 years from the sample to be zero, this finding may be due to a low sample size (only nine cases) for this age group.

The absolute change in LE between urban and rural areas for women was similar to that of men, but the gap in LE between urban and rural areas was greater for women of all ages. In contrast to men, although the absolute changes in the DemFLE for women in urban and rural areas still maintained positive growth, its growth was even greater than the increase in LE (except for women aged 75 and 85 years). Rural women will spend more time with dementia than urban women will. There are many employment opportunities and social resources in urban areas, and young people are more likely to be attracted to these areas. In China, as the children of rural elderly adults move into the city, their elderly parents may continue to live alone in the countryside, resulting in a large number of empty-nest elderly adults, especially rural women [7]. Particularly, as we mentioned earlier, at age 80, rural women seemed to perform surprisingly well. It was still found in this study that at age 80, the absolute number of years with dementia for rural women was lower than that for urban women.

The burden of dementia patients on themselves and society is a problem we must consider. At present, there is no medical treatment to change the disease, although several symptomatic medications exist with modest benefit on cognition [49]. Medical institutions need to invest substantial medical resources and formal care into the treatment of dementia patients. In addition, dementia patients undergo psychological and physiological damage, and informal caregivers also bear heavy economic and emotional burdens. Currently, the absolute number of people affected by dementia is increasing [5]. The medical management of dementia patients lags far behind the increasing burden caused by dementia. In the case of scarce medical resources and trained dementia doctors, [50] health departments need to carefully implement medical treatment, prevention and scientific research. In Jiangxi Province, more attention should be paid to urban men and rural women, and they should be given a proper increase in the allocation of resources. The Chinese government has introduced new policies in recent years, such as greater care for people over 65 [51]. In addition, the government has emphasized the role of rural doctors, which could lead to better care for dementia patients in rural areas [52]. However, these measures are not enough. Government decision-makers need to make reasonable prevention and management policies for the current severe situation. From the perspective of public health, the most important step is to implement a range of reasonable and effective primary prevention and secondary prevention strategies, [53,54,55] especially in rural areas. In other words, the government should work to reduce risks, increase population cognitive reserves, early detection and screening of diseased individuals. This is a cost-effective strategic measure that can play a huge role in the occurrence of late-stage dementia. For some elderly people, this may prevent them from suffering from dementia, or it may delay the time of suffering from dementia. Additionally, the entire elderly population should consciously be aware of the risk factors for dementia to prevent the occurrence of disease. In addition, even in older age, individuals should be actively involved in intellectual activities, which may help delay or prevent dementia [56].

Finally, some limitations should be addressed in this study. First, in previous reports, subjects may have moderate and severe dementia due to the problematic estimation of the prevalence of mild dementia [57]. In this study, the questionnaire data were the only source of the dementia status of the elderly participants. We were unable to make a clinical diagnosis of the subjects to determine their dementia levels, and we did not distinguish the specific types of dementia, such as vascular dementia and Alzheimer’s disease. Therefore, the prevalence of dementia we calculated will be higher, which may lead to our findings being higher than the actual situation. Second, due to the limited sample size of some age groups (≥95 years old), we had to merge these age groups. Third, in this study, only the differences between individuals in urban and rural areas were discussed. Some of the findings are not reasonably explained. For example, in urban areas, women were less affected by dementia than men. In addition, at age 80, rural women performed surprisingly well. Further research is needed regarding the risk factors for dementia-free life expectancy in Jiangxi Province.

## 5. Conclusions

With increased life expectancy, dementia-free life expectancy also increases, but the quality of life of the elderly population has not increased. Elderly urban men and elderly rural women experience more absolute years of dementia and a lower quality of life. These individuals need more attention and care from policy-makers and health workers.

## Figures and Tables

**Figure 1 ijerph-17-05665-f001:**
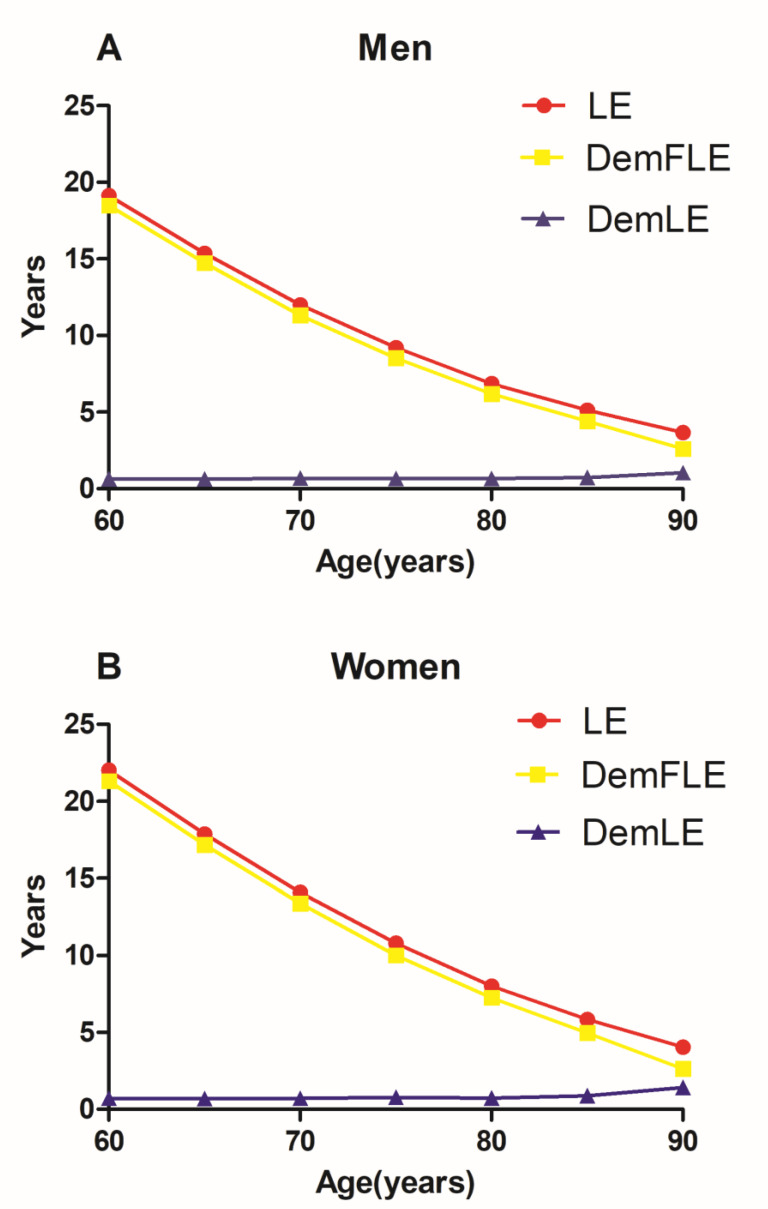
Trends of life expectancy (LE), dementia-free life expectancy (DemFLE), and life expectancy with dementia (DemLE). **A**. Men **B**. Women.

**Figure 2 ijerph-17-05665-f002:**
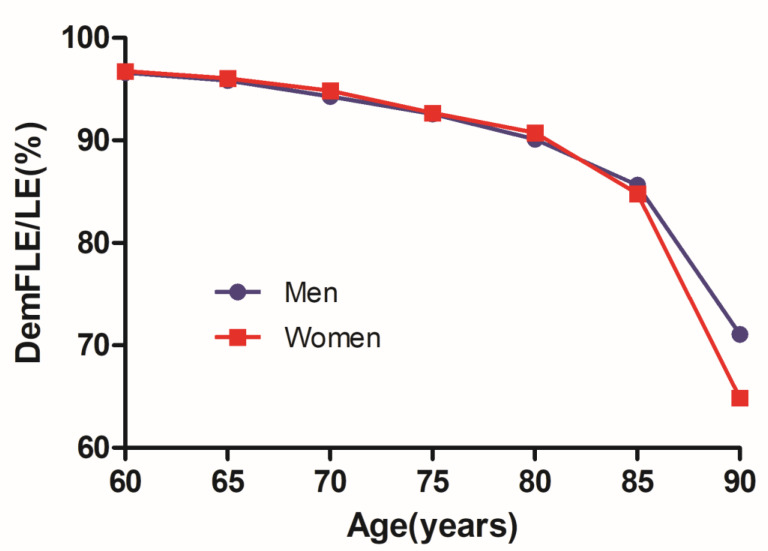
Trends of dementia-free life expectancy as a proportion of life expectancy (DemFLE/LE) for men and women.

**Figure 3 ijerph-17-05665-f003:**
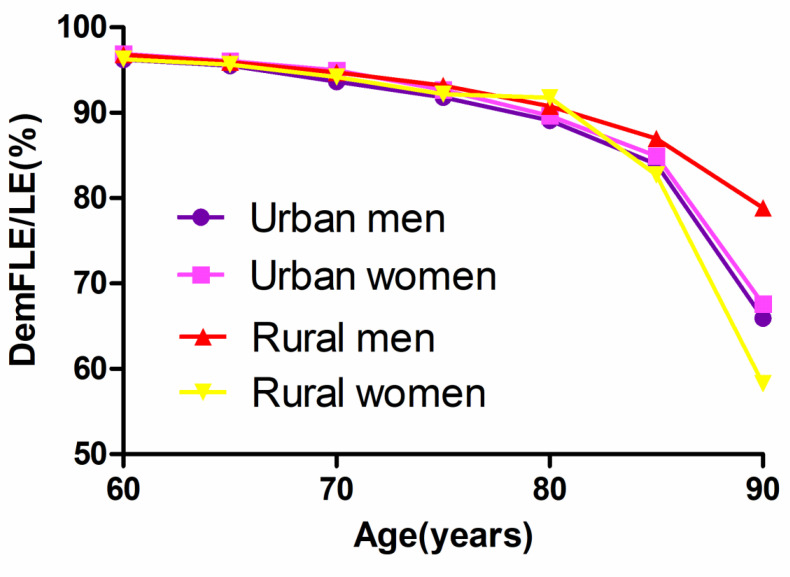
Trends of dementia-free life expectancy as a proportion of life expectancy (DemFLE/LE) for men and women in urban and rural areas.

**Figure 4 ijerph-17-05665-f004:**
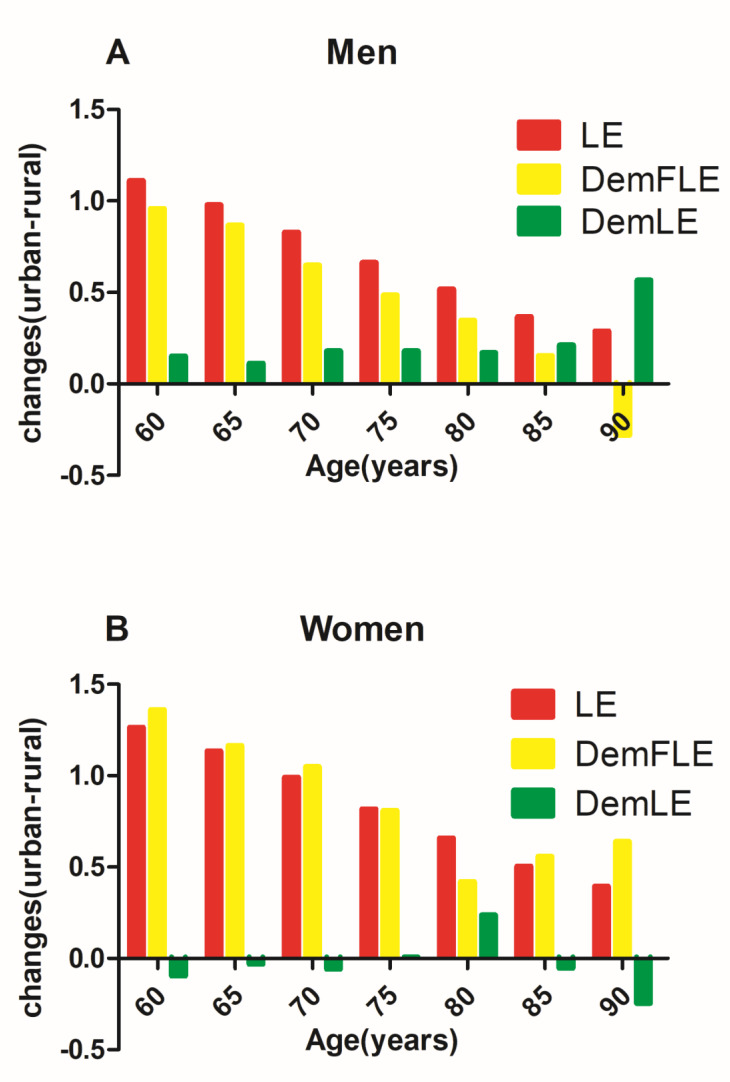
Absolute changes in life expectancy (LE) and dementia-free life expectancy (DemFLE) between urban and rural areas for men and women. **A**. Men **B**. Women.

**Table 1 ijerph-17-05665-t001:** LE, DemFLE, and the proportion of DemFLE in Jiangxi province, China, by sex and selected ages for the year 2018.

Age	Men				Women			
	*n*	Years	95% CI	DemFLE/LE (%)	*n*	Years	95% CI	DemFLE/LE (%)
60 years	268				275			
LE		19.13				22.03		
DemFLE		18.48	(18.31, 18.65)	96.62		21.31	(21.13, 21.50)	96.76
65 years	359				452			
LE		15.37				17.89		
DemFLE		14.73	(14.56, 14.91)	95.86		17.19	(17.00, 17.37)	96.05
70 years	272				248			
LE		12.02				14.10		
DemFLE		11.33	(11.14, 11.52)	94.28		13.38	(13.18, 13.57)	94.84
75 years	176				195			
LE		9.21				10.80		
DemFLE		8.53	(8.32, 8.74)	92.59		10.01	(9.79, 10.23)	92.68
80 years	137				134			
LE		6.86				8.01		
DemFLE		6.19	(5.96, 6.41)	90.11		7.27	(7.05, 7.49)	90.76
85 years	63				71			
LE		5.13				5.85		
DemFLE		4.40	(4.13, 4.67)	85.64		4.96	(4.70, 5.23)	84.82
90 years	32				29			
LE		3.67				4.05		
DemFLE		2.61	(2.32, 2.90)	71.08		2.63	(2.24, 3.01)	64.87

LE, Life expectancy; DemFLE, Dementia-free life expectancy; 95% CI, 95% confidence intervals.

**Table 2 ijerph-17-05665-t002:** LE, DemFLE, and the proportion of DemFLE for selected ages in urban and rural areas.

Age	*n*	Men			*n*	Women		
		LE	DemFLE (95% CI)	DemFLE/LE (%)		LE	DemFLE (95% CI)	DemFLE/LE (%)
Urban								
60	130	20.07	19.31(19.05, 19.57)	96.22	144	23.10	22.38(22.16, 22.61)	96.90
65	200	16.20	15.47(15.22, 15.72)	95.51	243	18.86	18.12(17.88, 18.35)	96.06
70	129	12.73	11.91(11.64, 12.19)	93.63	136	14.95	14.20(13.95, 14.44)	94.98
75	103	9.78	8.97(8.68, 9.26)	91.80	112	11.50	10.66(10.39, 10.93)	92.66
80	75	7.31	6.51(6.20, 6.82)	89.10	90	8.57	7.68(7.39, 7.97)	89.63
85	42	5.45	4.57(4.23, 4.92)	83.98	47	6.28	5.34(5.03, 5.64)	84.95
90	23	3.92	2.58(2.17, 2.99)	65.94	16	4.39	2.97(2.59, 3.34)	67.60
Rural								
60	138	18.96	18.35(18.10, 18.60)	96.80	131	21.84	21.02(20.72, 21.33)	96.28
65	159	15.22	14.61(14.34, 14.87)	95.95	209	17.72	16.95(16.65, 17.25)	95.64
70	143	11.90	11.27(10.98, 11.55)	94.70	112	13.96	13.15(12.83, 13.47)	94.20
75	73	9.11	8.49(8.17.8.80)	93.17	83	10.68	9.85(9.51, 10.19)	92.18
80	62	6.79	6.16(5.81, 6.51)	90.78	44	7.91	7.26(6.96, 7.56)	91.76
85	21	5.08	4.42(3.97, 4.87)	86.99	24	5.78	4.78(4.32, 5.24)	82.72
90	9	3.63	2.86(-)	78.89	13	3.99	2.33(1.63, 3.02)	58.26

LE, Life expectancy; DemFLE, Dementia-free life expectancy; 95% CI, 95% confidence intervals.

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
