# Peer review of "Dementia-Free Life Expectancy among People over 60 Years Old by Sex, Urban and Rural Areas in Jiangxi Province, China"

_ijerph, 2020, doi:10.3390/ijerph17165665_

Round 1
Reviewer 1 Report
Thank you for possibility to read the manuscript and learn more about your study. I found study topic significant and interesting. I think there was good reasoning about topic significance in China context in comparison with global situation. I do not have high competence on statistical methods used but they seems to be appropriate within research topic, also the limitations of study seems to be properly discussed. I have few specific comments and suggestions which could generally improve the manuscript.
Specific comments:
Abstract- conclusion section indicate most important trend from findings but practical implications are stated too general.
Introduction- in lines 62-63 as part of aim stated investigation on quality of life (QoL) but study had no specific findings on QoL.
Results- it seems that in Figure 1 information repeats (visually represents) the information in Table 1 and I would suggest to skip it.
Discussion- practical implications mainly refer that in Jiangxi Province, more attention should be paid to urban men and rural women, and they should be given a proper increase in the allocation of resources. Despite that few statements on current policies and their flaws are mentioned, more emphasis on situation complexity is lucking- I would recommend to highlight the public health perspective and need to increase of awareness in society, particularly among elderly population, on prevention strategies.
Author Response
尊敬的审稿人:
谨代表我的共同作者,非常感谢您给我们提供了修改稿件的机会,也非常感谢您对我们题为“ 60岁以上人群的无痴呆症的预期寿命”的稿件提出了积极和建设性的意见和建议。中国江西省城市和农村的性别年龄”。(ID:ijerph-875491)。
我们仔细研究了审阅者的意见,并进行了修订,并在论文中用红色标记了。此外,我们还提供了附件,以更好地回复您的评论,请参阅附件。我们已根据评论尽力修订稿件。
我们想对您和审稿人对我们的论文发表的评论表示由衷的感谢。期待您的回音。
感谢你并致以真诚的问候。
你的
真诚地
吴宇航

Reviewer 2 Report
Reviewer’s comments
The article is devoted to the study of urgent problems of social health during aging. Dementia is a widespread socially significant disease and the incidence of dementia is constantly increasing. The study of Dementia-free life expectancy is conducted in different countries, and the revised manuscript shows the results of such a study, first conducted in one of the provinces of China. The work considers not only the dependence of the development of dementia on age and gender, but also on the conditions in which the study participants live (urban or rural). The study was carried out according to the standard scheme adopted when performing similar studies in other countries and the text of the manuscript was compiled similarly to published articles on such topics. The manuscript is written using good English. However, there are a few comments.
Broad comments
line 17: LE and DemFLE were higher for those in urban areas than in rural areas, higher in women than in men. – According to the data in Table 2, this is true not for all age categories. It should be written more accurately.
Lines 21-22: Conclusions: With increased LE, DemFLE also increases, but with older age and over time, DemFLE/LE gradually decreases. The effect of dementia on elderly adults becomes more serious. – If the authors here mean the data in Table 2 on the DemFLE / LE difference expressed as a percentage (DemFLE / LE decreases to 71.08% for men and 64.87% for women at the age of 90), then they must prove that this difference is not accidental, but statistically significant, which is not shown in the work.
Lines 74-77: We used information on the infant mortality rate (IMR) and the under-five mortality rate (U5MR) in 2018 by gender and urban-rural areas, and the data come from the maternal and child monitoring system. On the basis of these data, we used the China model life table method [41] to estimate age-specific mortality rates and life expectancy by sex and urban and rural areas in Jiangxi Province in 2018. – it is not really understandable how it is possible to estimate the age-specific mortality rates and life expectancy for old people, based on data on mortality of children under 5 years old.
Lines 111-113: According to the text presented, the reader should understand that people aged 90 years have a difference in LE and DemLE between men and women. However, judging by the data given in Table 1 (data on the confidence interval), there is no difference between men and women at this age, respectively, and the DemFLE / LE difference expressed as a percentage (DemFLE / LE decreases to 71.08% for men and 64.87 % for women at the age of 90), cannot indicate differences.
The data for Table 2: why factor analysis was not used for data processing?
Specific comments
Line 112: The LE for women is 4.028 years – in the Table 1 there is a little different figure - 4.048 . Please, correct.
Tables 1 and 2: The number of subjects in each group is not indicated.
Author Response

(The authors gave the same response as above.)

Reviewer 3 Report
This is an interesting and useful paper – interesting in itself and useful in the mechanism used to generate the findings. It is quite thorough in its approach and is mostly well-written. However, there are some points where the authors could clarify their meaning - I have noted below where I think the text could benefit. More importantly, the authors could re-examine the discussion with a view to making somewhat shorter (and simpler).
I would recommend it be published with minor amendments, see below.
Page 1 line 58: Sentence beginning ‘However, there is evidence.. ‘ is a little clumsy, and.. might be better as something like ‘However, because of evidence [38, 39] that both sex and urban-rural differences are critical in chronic noncommunicable disease (NCDs) prevalence, our analysis of urban-rural differences is sex-specific.’
Page 2: line 65: the sentence beginning ‘It is hoped that the present study ..’ reads as if it refers to a study other than the present study – it might be easier if the authors stated it as something like .. ‘We hope this study will provide a reference point for similar studies elsewhere with the aim of establishing a basis for both domestic and international comparisons.’
Page 2: line 91: the response rate for the dementia survey was about 27% - could the authors say something about the response rate, for example how it compares with other similar surveys etc.
Page 3: line 104: in great detail redundant
Table 1: two decimal places is usually sufficient in the presentation of results.
Also it might be useful to also describe the derived proportions conceptually – ie as that proportion of the life expectancy that can be expected to be free of dementia. However, I’m not really convinced of its utility when a (simpler) absolute duration can be stated.
Page 6 line 150: I’m not sure I follow the logic at this point – could the authors please clarify their explanation.
Discussion: this could be shortened quite a bit without loss of clarity. Currently it feels there is some redundancy and repetition, which the authors should address.
Author Response
Dear Reviewer,
On behalf of my co-authors, we thank you very much for giving us an opportunity to revise our manuscript, we appreciate you very much for your positive and constructive comments and suggestions on our manuscript entitled “Dementia-free life expectancy among people over 60 years old by sex, urban and rural areas in Jiangxi Province, China”. (ID: ijerph-875491).
We have studied reviewer’s comments carefully and have made revision which marked in red in the paper. In addition, we have also provided the attachment for better reply to your comments,Please see the attachment. We have tried our best to revise our manuscript according to the comments.
We would like to express our great appreciation to you and reviewers for comments on our paper. Looking forward to hearing from you.
Thank you and best regards.
yours
sincerely
Yuhang Wu

Round 2
Reviewer 2 Report
The authors made the necessary changes to the text of the manuscript. I have no further comments.